# Piezoresistive Porous Composites with Triply Periodic Minimal Surface Structures Prepared by Self-Resistance Electric Heating and 3D Printing

**DOI:** 10.3390/s24072184

**Published:** 2024-03-28

**Authors:** Ke Peng, Tianyu Yu, Pan Wu, Mingjun Chen

**Affiliations:** 1State Key Laboratory of Robotics and System, Harbin Institute of Technology, Harbin 150001, China; pengke190929@163.com (K.P.); chnscwp@163.com (P.W.); chenmj@hit.edu.cn (M.C.); 2Key Laboratory of Micro-Systems and Micro-Structures Manufacturing, Ministry of Education, Harbin 150001, China

**Keywords:** 3D printing, triply periodic minimal surface, composites, piezoresistive, sensors

## Abstract

Three-dimensional flexible piezoresistive porous sensors are of interest in health diagnosis and wearable devices. In this study, conductive porous sensors with complex triply periodic minimal surface (TPMS) structures were fabricated using the 3D printed sacrificial mold and enhancement of MWCNTs. A new curing routine by the self-resistance electric heating was implemented. The porous sensors were designed with different pore sizes and unit cell types of the TPMS (Diamond (D), Gyroid (G), and I-WP (I)). The impact of pore characteristics and the hybrid fabrication technique on the compressive properties and piezoresistive response of the developed porous sensors was studied. The results indicate that the porous sensors cured by the self-resistance electric heating could render a uniform temperature distribution in the composites and reduce the voids in the walls, exhibiting a higher elastic modulus and a better piezoresistive response. Among these specimens, the specimen with the D-based structure cured by self-resistance electric heating showed the highest responsive strain (61%), with a corresponding resistance response value of 0.97, which increased by 10.26% compared to the specimen heated by the external heat sources. This study provides a new perspective on design and fabrication of porous materials with piezoresistive functionalities, particularly in the realm of flexible and portable piezoresistive sensors.

## 1. Introduction

Flexible pressure sensors have been increasingly used in wearable devices, electronic skins, and personal health monitoring [1,2,3], which are capable of converting pressure into corresponding electrical signals. Among the four types of pressure sensors (piezoresistive, capacitive, piezoelectric, and triboelectric [4,5,6,7]), piezoresistive pressure sensors comprised electrically conductive networks within an insulated matrix, and they arouse tremendous attention with a feasible fabrication process, high reliability, and low cost; these properties enable them to be used in the structural health monitoring for damage sensing and strain gauges [8,9]. Piezoresistive sensors can be classified into one-dimensional, two-dimensional, and three-dimensional structures. The 3D structure provides a larger pressure compression space, resulting in a stable resistance signal over a wider pressure range. With the rapid development of flexible piezoresistive pressure sensors, the structure design and manufacturing has been mainly focused [10]. Owing to the advantages of lightweight, large, specific surface area and high porosity, 3D porous structure has become an effective strategy to achieve promising piezoresistive performance, which has been applied in widespread fields [11].

It has been demonstrated that 3D TPMS structure used to design the structure of the piezoresistive pressure sensors with local area minimization is able to reduce both surface tension and surface energy, and lower residual stress [12]. Several 3D piezoresistive sensors have been realized using TPMS structure. For example, Peng et al. [13] prepared a flexible sensor with TPMS structure which exhibited lightweight air permeability and high stretchability. In addition, the sensors with TPMS structure exhibit higher resilience than those with bulk structure. Imanian et al. [14] engineered soft piezoresistive wearable conductors with TPMS-based architectures and evaluated the effects of pore shape on piezoresistivity in four different TPMS structures (i.e., Primitive, Diamond, Gyroid, and I-WP). Davoodi et al. [15] fabricated durable and flexible 3D conductive sensors with interconnected TPMS structures, and found that different structural cell types could result in different gauge factors. Ronca et al. [16] fabricated piezoresistive sensors using a selective laser sintering technique from graphene-wrapped thermoplastic polyurethane powders and investigated the electrical and thermal conductivity, as well as the mechanical strength of the porous structure designed by TMPS equations corresponding to Schwarz, Diamond, and Gyroid unit cells. TPMS structures have been used as synthetic bone grafts, confirming high permeability and great potential for bone repairing [17,18].

Different additive manufacturing technologies have been used for 3D piezoresistive pressure sensors. Wei et al. [19] fabricated 3D multifunctional polyurethane/carbon black sensors by direct ink writing using a 3D printer equipped with a 400 μm nozzle, and evaluated the effect of the ink composition on viscosity and printability. Kim et al. [20] fabricated a 3D multiaxial force sensor using fused deposition modeling 3D printing of carbon nanotube nanocomposite, and developed a simultaneous resistance measurement system for a real-time force sensing in three axes. Hohimer et al. [21] fabricated soft pneumatic actuators with piezoresistive sensing capabilities using multi-material fused filament fabrication. It was found that TPU-MWCNT could perform well as capacitive touch sensors in a piezoresistive flex mode under a variety of MWCNT loading conditions. Saadi et al. [22] developed a nanocomposite resin with low viscosity for vat photopolymerization 3D printing and investigated the piezoresistive and mechanical behavior of nanocomposites. Due to the excellent printability of the developed resin, piezoresistive pressure sensors with different regimes of deformation were realized. Fekiri et al. [23] demonstrated that the flexible, high-sensitivity piezoresistive sensors fabricated using material extrusion showed low hysteresis error and non-linearity error with an immediate response. However, sensitive soft materials such as polydimethylsiloxane (PDMS) are extremely difficult to be 3D printed due to the high viscosity of the extruded materials from the printing nozzle. To solve this issue, 3D printing technique is expected to be taken advantage of to build a sacrificial mold into which the conductive polymer is poured, and then the mold can be disposed of in various ways. Alsharari et al. [24] presented a method to fabricate soft compressible multilayer pressure sensors via sacrificial FDM 3D printing, which demonstrate a linear and reproducible response with wide range sensitivity. Sixt et al. [25] fabricated flexible piezoresistive pressure sensors by dissolving an acrylonitrile butadiene styrene sacrificial mold fabricated by FDM in acetone. Peng et al. [26] fabricated a porous flexible strain sensor by casting polyurethane/carbon nanotube composites into the DLP-printed sacrificial mold, which showed high stretchability and excellent recoverability.

In general, conductive polymer composites are usually heated at a low temperature for a long time by the external heat source with the furnace during the fabrication process. The conductive fillers are usually dispersed into the elastomer matrix to make the sensors highly conductive. With the electrical conductivity and the Joule heating effect of fillers, the self-resistance electric heating is of great interest, given that it has shorter preparation time, lower energy consumption, and higher mold dimensional stability compared to using the external heat source [27,28,29]. Zhang et al. [30] investigated the mechanical properties of carbon fiber reinforced epoxy laminates fabricated by the self-resistance electric heating. They found that the content of carbon nanotubes addition could improve the thermoset curing performance and mechanical properties of the composites. Collinson et al. [31] cured carbon fiber reinforced polymer (CFRP) composites with the self-resistance electric heating and compared it to the oven curing process. The results showed that the average void content of the samples was lower by 0.82% compared to the oven-cured samples, and the energy consumption was significantly reduced. Liu et al. [32] developed a multiple zone self-resistance electric heating method, which reduced the overall cure-induced distortion of the CFRP part. Compared to the traditional oven curing, the multiple zone self-resistance electric curing process achieved an average reduction of 67.46% in the curing-induced distortion.

However, little research has been reported on pressure sensors with a porous TPMS structure generated using the self-resistance electric heating with a 3D printed sacrificial mold. The introduction of new pore geometries by 3D printing, as well as the self-resistance electric heating, are critical to the mechanical and piezoresistive properties of the sensors. This paper aims to study porous piezoresistive sensors composed of PDMS and multi-walled carbon nanotubes (MWCNTs) based on 3D TPMS structures with the self-resistance electric heating. Three different electric fields were applied to the conductive composites during the heating process. The compressive and piezoresistive properties of the porous composites based on different TPMS types such as Diamond, Gyroid, and I-WP unit cells were studied. By filling the research gap and introducing new approaches such as TPMS-based pore geometries and self-resistance electric heating, this study has the potential to enhance the design and fabrication of next-generation pressure sensors for various applications including wearable devices, electronic skins, and health monitoring systems.

## 2. Methods and Materials

### 2.1. Design of Sacrificial Models Based on TPMS Structures

For the design of 3D porous structures, TPMS has been used. TPMS has a minimal surface periodic in three independent directions, extends infinitely, and in the absence of self-intersections, divides the space into two labyrinths. TPMS can be obtained using trigonometric functions with three-dimensional architectures. Trigonometric functions that described the architectures of Diamond (D), Gyroid (G), and I-WP (I) are as follows:(1)D: sin(x)sin(y)sin(z)+sin(x)cos(y)cos(z)+cos(x)sin(y)cos(z)+cos(x)cos(y)sin(z)=C 
(2)G: sin(x)cos(y)+sin(y)cos(z)+sin(z)cos(x)=C
(3)$$I:\;2\left(\cos(x)\cos(y)+\cos(y)\cos(z)+\cos(z)\cos(x)\right)-(\cos(2x)+\cos(2y)+\cos(2z))=C\;$$
where X = 2απ*x*, Y = 2βπ*y*, and Z = 2γπ*z*. The left-hand side of the equations controls the iso-surface shape, and the parameter C on the right-hand side represents the iso-value at which the surface is plotted. Here, the coefficients α, β, and γ are used to control the dimensions of the unit cell along the *x*, *y,* and *z* directions. The parameter C is the offset which controls the porosity of the structures. While the derived structure is uniform with the constant C, the derived structure is gradient with the field variable C. In this study, the minimal surface generation software MSLattice V1.0 is used to generate the TPMS structure with solid networks. The TPMS structure, namely Diamond (D), Gyroid (G), and I-WP (I), with a single unit cell as shown in Figure 1. Relative density is one of the most important properties for a lattice structure. Three relative density values (0.2, 0.3, and 0.4) were set for each geometry in order to study the effect of porosity on mechanical behavior and electrical conductivity. The original CAD models consisted of cuboidal porous geometries with a height of 20 mm and side length of 10 mm where 4 × 2 × 2 unit cells were repeated along with a 5 mm cubic unit cell, as shown in Figure 1. The relative densities of the remaining conductive structures after dissolution of the sacrificial samples were 0.8, 0.7, and 0.6, respectively. In the following, Diamond, Gyroid, and I-WP based porous structures are labeled as DM, GM, and IM, respectively, whereas D, G, or I represent the geometry and M represents the density value. For example, G0.8 represents a Gyroid-based architecture with a relative density of 0.8.

### 2.2. Preparation of Conductive Porous Structures

Acryloyl morpholine (ACMO) was used as the monomer and was obtained from KJ Chemicals Co., Tokyo, Japan. Diphenyl (acyl) phosphine oxide (TPO) was used as the photoinitiator and Sudan Orange G was used as the light absorber, and both were purchased from Aladdin Co., Ltd., Shanghai, China. 1-decanethiol (purity: 96%) was used as the free radical chain transfer agent and polydimethylsiloxane (PDMS) and its curing agent were obtained from Sigma Aldrich, Germany. Multi-walled carbon nanotubes (MWCNTs) (diameter 9.5 nm, length 1.5 μm, purity > 90%) were purchased from Shanghai Xiyan Co., Ltd., Shanghai, China. All materials were directly used without further purification.

The water-soluble sacrificial samples were fabricated using a liquid crystal display (LCD, vat photopolymerization based) 3D printer (ID-002H, Creality Co., Ltd., Huizhou, China). The main components of the water-soluble sacrificial resin were ACMO, TPO, 1-decanethiol, and Sudan Orange G. To prepare a sacrificial resin solution, 1 wt.% TPO, 0.5 wt.% 1-decanethiol, and 0.025 wt.% Sudan Orange G were added to the monomer ACMO solution, stirred at 500 rpm for 15 min to form a uniform mixture, and then placed at 70 °C for 15 min until the resin was homogenous. After designing the structure of sacrificial samples, the commercial software Chitubox V2.0 was used to slice and define the printing parameters with a UV light wavelength of 405 nm, an exposure time of 4 s for the bottom layer and 3.5 s for the rest layer, and a 500 μm layer thickness. After printing, the excess resin on the surface of the printed part was removed and the printed parts were post-cured in a UV chamber at 60 °C for 10 min.

The PDMS-MWCNTs composites solution consisted of PDMS, curing agent and MWCNTs and the weight ratio of PDMS and curing agent was 10:1. The appropriate amount of MWCNTs were first added to the acetone solution and ultrasonicated for 10 min to achieve effective dispersion of MWCNTs, then the PDMS solution was added and stirred at 500 rpm for 10 min, and after homogenous stirring, the mixed solution was stirred at 80 °C to evaporate the acetone solution. After the acetone solution evaporated, curing agent was added to the cooled mixed solution and stirred in vacuum mixer at 500 rpm for 15 min to obtain the thermosetting PDMS-MWCNTs matrix solution. Subsequently, the matrix solution was injected into the sacrificial mold, and a voltage provided by the power supply (HY-4000W-100, Qidong Bohai Electronics, Dongguan, China) was applied on both sides with a pair of copper plate electrodes to cure the matrix solution, while the control group was cured in a heating oven for 30 min at 100 °C. Then, the solid matrix composites were placed in the water at 50 °C for 10 h until the sacrificial mold was completely dissolved. By drying the remaining solid phases at 60 °C for 30 min, the flexible pressure sensors were obtained.

### 2.3. Measurement and Characterization

Static compression tests (standard: ASTM D1621) were carried out by using a mechanical testing machine (WDW-02, Shijing Corp., Jinan, China). A multimeter (Keithley 2450 digital multimeter, Tektronix, Beaverton, OR, USA) was used to measure the resistance of specimens. Two electrodes made of conductive copper tape were glued to the top and on the bottom of the specimen and connected with the multimeter through copper wires. Compression tests were performed while monitoring the change of resistance during loading and deformation. A schematic of the piezoresistivity measurements is shown in Figure 2. The mechanical properties were evaluated by subjecting the specimens to cyclic compression/strain up to 40% of the initial value of the cubic specimen length, at a deformation rate of 20 mm/min, at 25 °C. The electrical conductivity (σ) of the sensor was calculated according to the following formula: σ = L/RA, where R is ohmic resistance, L is the length, and A is the cross-sectional area of the sensor. The temperature during the heating process was tested using a thermal imager (HM-TPK20-3AQF/W, Hikmicro, Hangzhou, China).

## 3. Results and Discussion

### 3.1. Manufacturing of Porous TPMS Composites

Porous strain sensors made of PDMS-MWCNTs were fabricated using a 3D printed sacrificial mold to shape the PDMS into the desired geometry. The fabrication process is shown in Figure 3a. First, the sacrificial molds with different TPMS structures were printed by an LCD 3D printer, which served as templates for creating the desired porous structure in the PDMS-MWCNTs composites, as shown in Figure 3b. Then, the PDMS-MWCNTs composites solution was poured into a container with a LCD sacrificial mold. After the solution had completely penetrated the interconnected pores, the heating process was initiated. The traditional heating method involved heating in the furnace to 100 °C for 30 min, while the self-resistance electric heating method required placing a pair of copper electrodes with a voltage strength of 25 V/cm for 10 min on the inner wall of the regular container. The PDMS solution doped with MWCNTs exhibits electrical conductivity, which causes a rapid temperature increase and enables rapid curing. Figure 3d displays the solid PDMS-MWCNTs matrix phases after a full cure with various TPMS structures. After the complete curing process, the LCD sacrificial mold was dissolved thoroughly by immersing it in water, leaving behind the flexible and porous PDMS-MWCNT scaffold. By following this fabrication process, the researchers were able to create porous strain sensors with controlled geometries and properties, paving the way for further investigations into the mechanical and piezoresistive characteristics. The PDMS-MWCNTs scaffold successfully replicated the surface topology of the sacrificial mold, ensuring the transfer of TMPS structure features to the scaffold’s surface. This approach improves localized homogeneity and surface precision compared to direct printing methods. The accuracy of the LCD sacrificial molds was assessed by measuring porosity through dry weighing, as presented in Table 1. The measured sacrificial phase porosity values deviated by no more than 11.67% from the CAD porosity values. Additionally, the sensing performance of the PDMS-MWCNTs composites is affected by their electrical conductivity. Figure 3c illustrates the electrical conductivity of PDMS-MWCNTs composites with different MWCNTs content at the room temperature. The results indicate that an increase in MWCNT content could lead to a corresponding increase in conductivity. The conductivity rapidly increases by several orders of magnitude when the content reaches the percolation threshold (around 0.5 wt.%). The conductivity of the PDMS-MWCNTs composites with a mass fraction of 1.0 wt.% of MWCNTs was 1.10546 × 10^−2^ S/cm. However, further increasing the MWCNTs content could increase the brittleness and reduce the printability of the resins. Additionally, higher MWCNTs content increases the viscosity of the PDMS-MWCNTs composite solution, resulting in poor fluidity during the casting process and potential formation of defective edges in the porous material, as depicted in Figure 3f. To ensure optimal electrical conductivity and processing performance, a weight ratio of 1 wt.% was determined. The distribution of nanofiller in the polymer matrix is crucial for the piezoresistive material. Various methods have been developed, including extrusion filament with a corotating twin-screw extruder and pre-preg technology [33,34]. Ronca et al. dispersed graphene in ethanol, subjected it to ultrasonication, and mechanically stirred it with the polymer matrix [16]. In our study, MWCNTs were dispersed in an acetone solution and ultrasonicated for 10 min. The mixture was then mechanically stirred with PDMS at 80 °C to evaporate the acetone solution. To achieve good dispersion, the relationship between dispersion time and conductivity was explored, as shown in Figure 3e. The conductivity increased as the dispersion time increased. Specifically, the conductivity of the sample dispersed for 10 min was twice more than that of the sample dispersed for 5 min. However, the increase rate of conductivity slowed down for a further increase of the dispersion time. Considering both the dispersion time and conductivity, a dispersion time of 10 min was chosen in this study. Therefore, the PDMS-MWCNT composites with 1 wt.% of MWCNTs and a dispersion time of 10 min were used in the tests and material characterization. Figure 3g presents the tensile stress–strain curves of pure PDMS and PDMS-MWCNTs composites with 1 wt.% of MWCNTs. The addition of MWCNTs influenced the mechanical properties of PDMS, resulting in a higher tensile strength compared to pure PDMS. However, the difference in elongation at break between the two materials was only 1.8%.

### 3.2. Self-Resistance Electric Heating Process

This study aimed to compare the traditional furnace heating method, which involved placing the composite solution into a furnace at 100 °C for 30 min, with the self-resistance electric heating method. The latter method, without altering the content, made use of Joule heat generated by the self-resistance of MWCNTs in PDMS-MWCNTs composites to heat and cure for 10 min. A percolation network formed upon reaching a certain amount of MWCNT content, leading to an electro-thermal effect as the electric current passes through. Figure 4a illustrates how the internal heat source facilitates the heating and curing process of the composites. During the self-resistance electric heating process, parameters such as current and voltage influenced the temperature of the composite material. Thus, to regulate the temperature, the electric field strength at both sides of the copper electrode was adjusted, as shown in Figure 4b. The relationship between electric field strength and temperature change was found to be rapid, with the temperature of the composite solution increasing swiftly upon the application of the electric field. Moreover, a higher electric field strength resulted in a faster rate of temperature increase and a higher temperature within the same timeframe. To prevent violent chemical cross-linking reactions in the PDMS, an electric field strength of 25 V/cm was selected for heating, as indicated in the thermal imaging diagram in Figure 4d. The experiment demonstrated that the temperature between the electrodes increased rapidly after 5 s of applying the electric field, reaching 90 °C within 240 s. As the composite material dissipated heat, the rate of temperature rise slowed down, leading to reduced temperature differences within the material. The non-flat surface of the composite material, as shown in Figure 4c, resulted from the varying heat capacitance between the composite and electrode materials. However, uniform heat conduction among internal solutions led to minimal temperature variations and smooth internal pores within the prepared composite material, as depicted in Figure 4e.

### 3.3. Compressive Stress–Strain Behavior of Porous Composites

The mechanical behavior of PDMS-MWCNTs porous structures was examined through compression tests to assess the impact of relative density, processing technique, and geometry. The stress–strain curves for all samples are illustrated in Figure 5a–c. Initially, there was a linear elastic region, and the elastic modulus was determined from the linear section of the curves for all geometries considered, as shown in Figure 5d. The study revealed that the I-based structure, with relative densities of 0.6 and 0.7, displayed a plateau of nearly constant stress after the linear elastic region. This suggested that the I-based structure has a higher energy absorption capacity compared to the D- and G-based structures. During loading, the pore wall bent, leading to linear elasticity. Upon reaching a critical stress level, pore collapse occurred. At higher strains, either the pores collapsed or the walls made contact, causing further deformation. The steep increase in the stress–strain curve indicated compression within the pore wall material, signaling the onset of densification. Increasing the relative density of the porous composite material resulted in thicker pore walls, enhancing bending resistance and pore collapse stress. The analysis indicated that both the modulus and platform stress were elevated with the increase in relative density. Examining the stress–strain curves of the I-based structure revealed that the structure with a relative density of 0.6 exhibited a lower modulus and platform stress compared to the structure with a relative density of 0.7. Similarly, for the D-based structure, lower relative densities led to smaller strains at the initiation of densification. Reduced distances between pore walls in porous composites with lower relative densities facilitated quicker contact between pore walls, reducing the initial densification strain. The elastic modulus of the porous samples fabricated via self-resistance electric heating was slightly higher than that of the samples produced through traditional furnace heating. This increase in the elastic modulus of the cured porous complex structure can be attributed to the uniform current conduction and temperature distribution during the self-resistance electric heating process.

Figure 6 displays the stress–strain response curve of the porous composite material under cyclic compressive loading at various strains ranging from 10% to 40%. Within the low strain range (10% to 20%), the hysteresis zone was narrower. Upon unloading, the matrix phase exhibited near-complete strain recovery, indicating high resilience in the porous composite material. As the strain increases, there was a noticeable shift in the hysteresis loop. To precisely measure the extent of hysteresis in the high strain range (30% to 40%), we introduce the following hysteresis parameter:$$h=\frac{A_{unloading}-A_{loading}}{S_{loading}}$$
where *A* represents the area under the stress–strain curve. Figure 7a–c display the hysteresis parameters of porous composite materials with different structures. The findings indicated that higher strains are associated with increased hysteresis parameters, indicating a more significant hysteresis effect as the strain magnitude rises. Structures with higher relative density generally displayed larger hysteresis coefficients, underscoring the importance of the hysteresis phenomenon. Analyzing the hysteresis coefficients across various strains revealed notable variations in the I-based structures. Particularly, the strain recovery of the I-based structure was slow at high strains, reaching a hysteresis coefficient of 0.31 at a strain of 40%. In contrast, both D- and G-based structures exhibited the hysteresis coefficient within a relatively small range of 0 to 0.1. However, for the I-based structure, the hysteresis coefficient showed higher values as the relative density increases to 0.4, indicating a significant influence of relative density on the hysteresis performance of the I-based structure. The comparison of the hysteresis coefficients between the PDMS-MWCNTs scaffolds fabricated using these two methods revealed that samples cured by the traditional furnace heating method had smaller hysteresis coefficients. This suggests that the self-resistance electric heating method has the potential to partially mitigate the hysteresis within the structure.

Figure 7d,e depict the stress attenuation of composite materials after 50 compression cycles across different structures and processes, with the maximum stress values at 10% strain plotted as a function of cycle numbers. The results clearly show that regardless of geometry, all structures exhibited excellent stability and repeatability after the initial compression cycles. Among structures prepared using two different heating methods, the G-based structure demonstrated the highest maximum stress retention rate, followed by the D-based structure, while the I-based structure shows the lowest retention rate. This indicated that the fatigue effect of the G-based structure under cyclic loading is minimal. Moreover, the maximum stress retention rate of the model fabricated using the self-resistance electric heating process was notably higher than that of the model produced via the traditional heating method. From the analysis of the compressive performance of the structures fabricated using the self-resistance electric heating, it was clear that the compressive performance of the self-resistance electric heating cured samples is better than the traditional heating cured samples. In the process of self-resistance heating utilizing nano-enhancement such as MWCNTs, a notably high heating rate could be obtained at the interface between the fibers and the PDMS matrix. This elevated heating rate induces a temperature differential along the interface, leading to preferential heating of the substrate in the vicinity of the interface region. The resultant preferential heating phenomenon plays a crucial role in significantly improving the quality of the substrate as it cures, particularly enhancing the mechanical properties of the material in the interface region [35].

### 3.4. The Piezoresistive Behavior of Porous Composites 

The resistive response in this study is defined as the change in resistance (ΔR) relative to the initial resistance (R₀), expressed as ΔR/R₀ = (R₀ − R)/R₀, where R represents the resistance with compressive stress and R₀ represents the resistance without compressive stress. Figure 8a illustrates the results of the minimum response strains for the I-based structure with a relative density of 0.8. The resistance response showed a clear trend of increase, and decreased as the strain reached 1.5%, corresponding to its behavior. With an increase in strain to 2%, the resistance response became more prominent. For the I-based structure with relative densities of 0.6 and 0.7, the minimum response strains were 4% and 3%, respectively. This implies that structures with higher relative densities are more suitable for use as piezoresistors in detecting small deformations. The electromechanical cycling stability of TPMS structures made from PDMS-MWCNT composites was assessed. The samples underwent five consecutive compressive cycles (at 20% strain) at cycling rates ranging from 10 mm/min to 50 mm/min, as shown in Figure 8b. It indicated that the maximum response range does not exceed 11.95% at different cycling rates, demonstrating excellent stability and signal reversibility. At a compression rate of 50 mm/min, the resistance response value was significantly smaller than at 10 mm/min due to incomplete deformation restoration. Figure 8c presents the resistance response values of all structures with different relative densities at a strain of 10%. Among the G-based structures, the largest resistance response value was observed. Specifically, the structure with a relative density of 0.7 exhibited a larger resistance response value compared to the structure with a relative density of 0.6, attributed to the higher degree of densification and larger contact surface between the pore walls in the former. Consequently, the resistance value was smaller, leading to a larger resistance response value. However, when the relative density of the structure reached 0.8, it experienced a high degree of densification. As a result, increasing the strain had less impact on the resistance, resulting in a smaller resistance response compared to a structure with a relative density of 0.7. All samples in the study exhibited negative piezoresistive behavior, indicating that their electrical resistance decreased with increasing strain. Figure 8d illustrates the resistance response of the D0.8 structure, suggesting that when the compressive strain exceeds 20%, the samples’ internal lattice undergoes compaction and transitions from structural deformation to composite deformation. This led to a slower increase in the resistance response value, which stabilized around 0.8 when the strain reaches 30%. Figure 8e compares the responsive strain of structures prepared using traditional heating and self-resistance electric heating. The results indicated that the maximum response strain of the self-resistance electric heated structures was greater than that of the traditionally heated structures. Specifically, the structure prepared using the self-resistance electric heating process exhibited a larger maximum response strain and corresponding resistance response value compared to structures prepared using other processes. The D-based structure demonstrated the highest response strain and the corresponding resistance value among all structures. This suggests that the D structure with a density of 0.7 is suitable for use within a large strain range. The relationship between the compression load and the resistance was illustrated in Figure 8f. The pressure sensitivity (S), represented by the slope of ΔR/R_0_ against compression load (P), is a critical metric for assessing sensor performance. Two distinct plateaus in the D0.7 sample indicated two different compression stages. For the compression load under 0.068MPa, sensor deformation was predominantly influenced by the porous structure. For compression load > 0.068 MPa, the pores underwent significant compaction, enhancing the conductive path and subsequently elevating pressure sensitivity (S_2_ = 9.71 MPa^−1^).

## 4. Conclusions

This paper presents a feasible and efficient fabrication process for TPMS porous structures that exhibit improved mechanical recovery. This study utilizes MWCNT-doped composites’ conductivity, and employs self-resistive electric heating to prepare uniformly heated conductive materials. Piezoresistive sensors of PDMS with complex internal pores were fabricated based on 3D printed sacrificial models. The key conclusions are drawn as follows:Good energy absorption capacity and a high elastic modulus were achieved by the composites with serious hysteresis and larger stress loss in repeated cycles. Self-resistance electric heated samples exhibit a larger elastic modulus, a smaller hysteresis coefficient, and good retention of maximum strain in cyclic tests, indicating better compressive properties compared to externally heat-cured samples.The piezoresistive response was influenced by the relative density and structure of the samples. Among different structures, the D-based structure demonstrated the highest responsive strain of 61%, accompanied by a resistance response value of 0.97. The D-based structure sensors with a relative density of 0.7 showed the optimal resistance response within a large range of strain.Self-resistance electric heating samples exhibit a larger strain range and resistance response value compared to samples cured by the traditional furnace heating method, emphasizing the importance of selecting the appropriate relative density and structure based on the strain range used.

In conclusion, by selecting an appropriate relative density and structure, utilizing the self-resistive electric heating technique, sensors with good strain response could be prepared. This research provides valuable insights into the design and optimization of personalized piezoresistive sensors using 3D printing and molding.

## Figures and Tables

**Figure 1 sensors-24-02184-f001:**
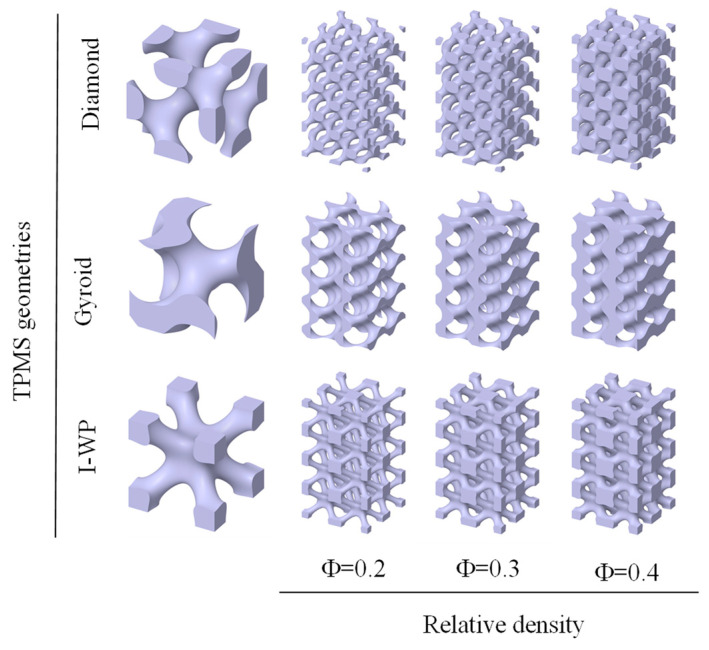
TMPS structure used as sacrificial models.

**Figure 2 sensors-24-02184-f002:**
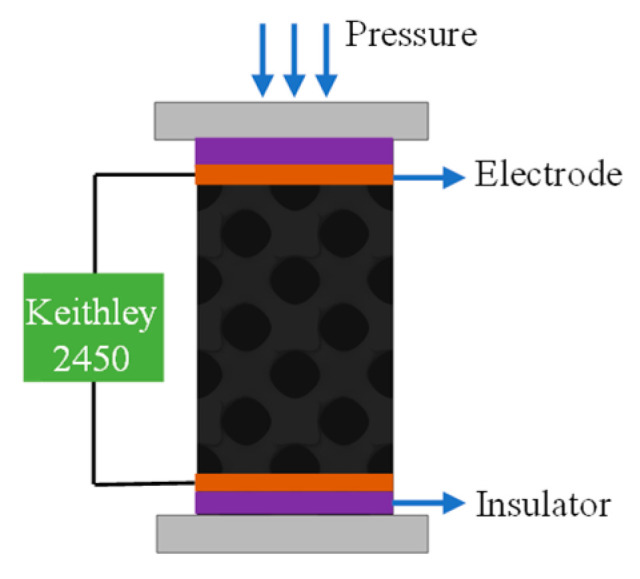
Schematic of the piezoresistivity measurement for the developed porous sensor.

**Figure 3 sensors-24-02184-f003:**
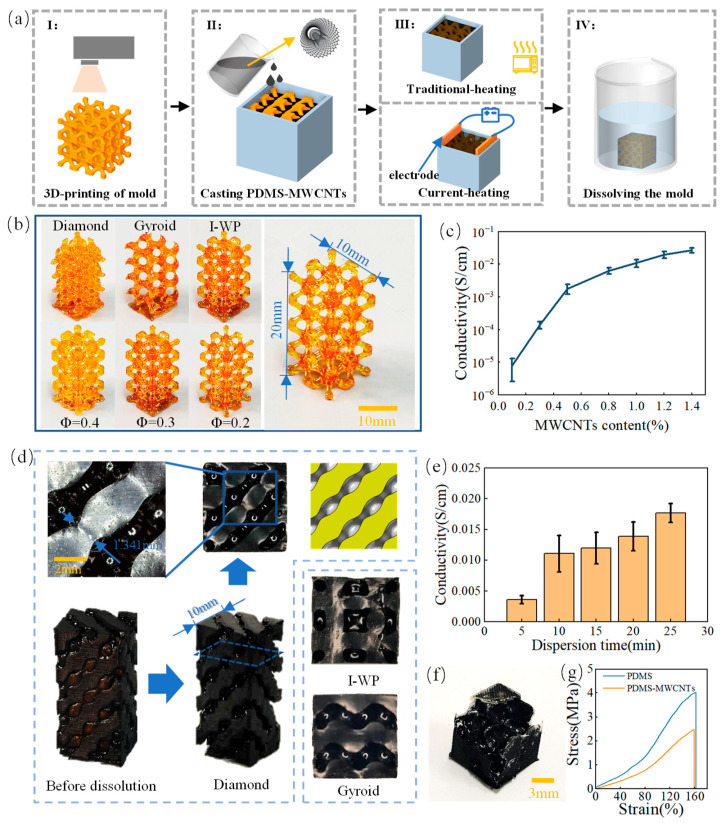
Fabrication of PDMS-MWCNTs porous composites. (**a**) 3D printing of sacrificial molds and preparation of the porous composites; (**b**) Sacrificial molds; (**c**) Conductivity of PDMS-MWCNTs composites with different MWCNTs content; (**d**) Porous composites with different TPMS structures; (**e**) Conductivity of PDMS-MWCNTs composites with different dispersion time; (**f**) Defective samples with the excessive content of MWCNTs; and (**g**) Tensile curves of pure PDMS and PDMS-MWCNTs composites.

**Figure 4 sensors-24-02184-f004:**
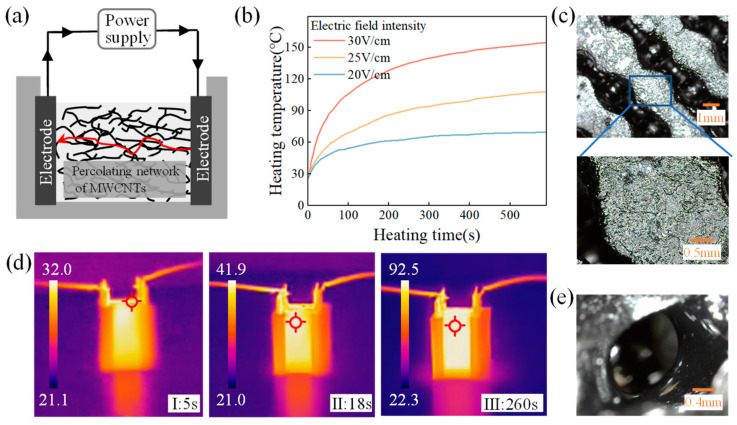
Self-resistance electric heating process. (**a**) Schematic of self-resistance electric heating; (**b**) The temperature during the heating process; (**c**) Surface of the composite in contact with the electrodes; (**d**) Thermal imaging images during the heating process; and (**e**) Smooth micro-pores inside the composites.

**Figure 5 sensors-24-02184-f005:**
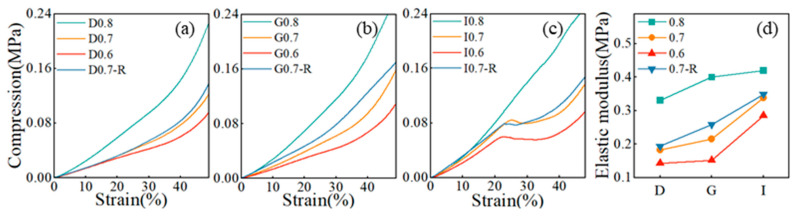
Compressive mechanical properties of porous matrix phases. (**a**–**c**) Stress–strain curves of porous matrix phases; (**d**) Elastic modulus of porous matrix phase (the labeled R stands for porous sensors fabricated by the self-resistance electric heating process, and the rest represent porous sensors cured by the furnace).

**Figure 6 sensors-24-02184-f006:**
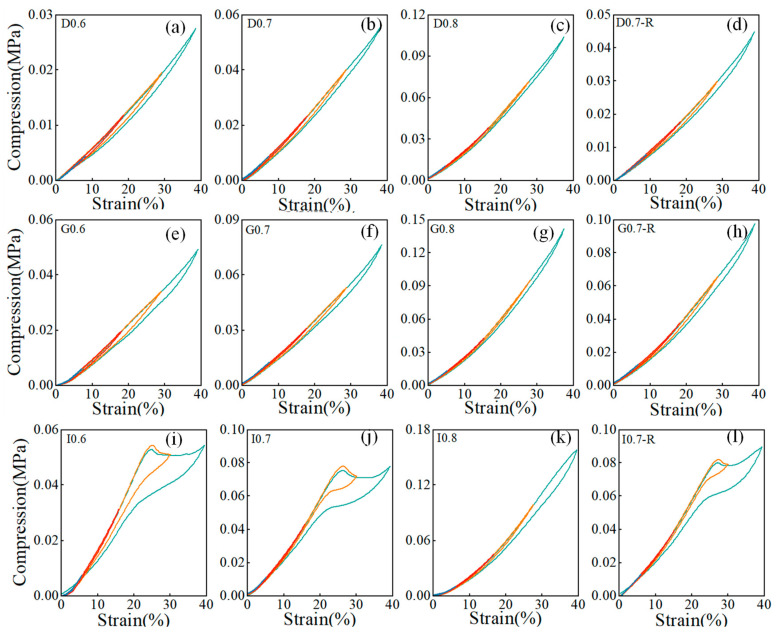
Cyclic compressive stress–strain curve under different strains. (**a**–**l**) Cyclic compressive stress–strain curves of sample D0.6, D0.7, D0.8, D0.7-R, G0.6, G0.7, G0.8, G0.7-R, I0.6, I0.7, I0.8 and I0.7-R.

**Figure 7 sensors-24-02184-f007:**
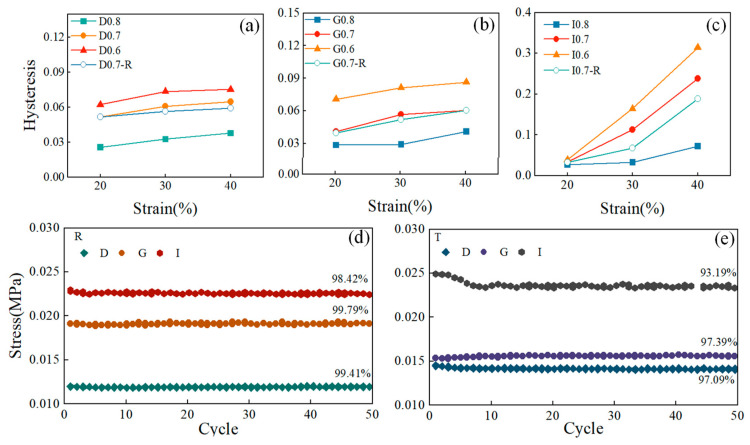
Cyclic compression performance. (**a**–**c**) Hysteresis parameters under different strains; (**d**,**e**) Maximum strains in the cyclic strain with different heating methods.

**Figure 8 sensors-24-02184-f008:**
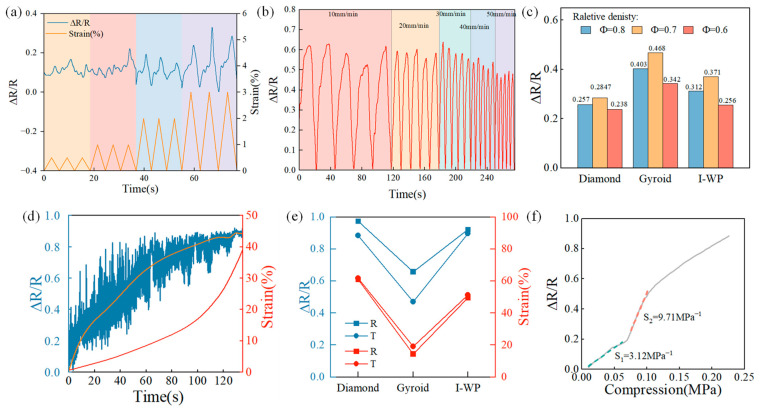
The piezoresistive behavior of porous composites. (**a**) The minimum response strains of the I0.8 sample; (**b**) The resistance response of the D0.8 sample; (**c**) The resistance response values of all structures with different relative density at a strain of 10%; (**d**) The resistance response of the D0.8 sample; (**e**) The responsive strain corresponding to resistance response value; and (**f**) The pressure sensitivity of the D0.7 sample.

**Table 1 sensors-24-02184-t001:** Porosity table of experimental samples and CAD design samples.

TMPS Type	Design Relative Density	Measured from the Fabricated Part				
		Height (mm)	Width (mm)	Depth (mm)	Relative Density	Error (%)
Diamond	0.2	10.39	10.49	19.8	0.21	6.77
	0.3	10.59	10.47	20.08	0.31	4.48
	0.4	10.45	10.52	19.78	0.41	1.67
Gyroid	0.2	10.45	10.47	19.77	0.21	4.18
	0.3	10.63	10.63	20.07	0.31	4.51
	0.4	10.52	10.49	19.77	0.42	4.07
I-WP	0.2	10.49	10.49	19.91	0.22	11.67
	0.3	10.56	10.56	19.90	0.32	8.20
	0.4	10.48	10.48	19.92	0.43	6.60

## Data Availability

Data is available upon request.

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
