# Peer review of "Piezoresistive Porous Composites with Triply Periodic Minimal Surface Structures Prepared by Self-Resistance Electric Heating and 3D Printing"

_sensors, 2024, doi:10.3390/s24072184_

Round 1

Reviewer 1 Report

Comments and Suggestions for Authors

You should include more studies on the use of additive manufacturing for 3D piezoresistive fabrication in your literature review, as well as add justifications for the technology you employed.

You should include the clear geometry of a 3D-printed sacrificial mold.

It is recommended to include real images for various TPMS structures.

It would be best if you added more details about the 3D LCD  printing parameters.

In the conclusion, you should specify the best TMPS type and density.

It is preferable to add details about data acquisition and the measuring system.

After selecting the best model, you should determine the transfer function between input and output.

Author Response

The authors appreciate the constructive comments and suggestions made by the reviewers. We have made corresponding modifications based on your comments, please see the attachment. Thank you a lot.

Reviewer 2 Report

Comments and Suggestions for Authors

128 - I don't understand what model of 3D printer is used for this research.

163 - Which software is used for testing of mechanical properties of material?

165 - Is this deformation rate or testing speed from the testing machine?

255 - Which standard is used for compression testing?

257 - Is the extensometer used for calculating elastic modulus?

263 - How stress was calculated when the cross-section surface of the specimen was not solid?

Author Response

(The authors gave the same response as above.)

Reviewer 3 Report

Comments and Suggestions for Authors

This study provides valuable insights into the piesoresistive flexible sensors prepared by Joule Heating and 3D printing. The paper needs major revision before acceptance for publication. The following are my comments:

1.       The abstract does not clearly explain the objectives of the study.

2.       How has the fabrication method enhanced the compression performance and piezoresistive response of the nanocomposite materials? Please write a short sentence about this in the abstract.

3.       What is the advantage of joule heating to the traditional heating method such as curing in oven in terms of mechanical and electrical properties of prepared piesoresistive material?

4.       The authors mentioned the four types of pressure sensor and specifically peisoresistive sensor. How and why piesoresitive sensors have made? And why they are important? For example you can say they are comprised of a conductive networks within an insulated matrix. Their properties enables them to be used in Structural health monitoring for damage sensing and for strain gauges. In this regards, there are a lot of papers as the following which can be indicated here.

[a] Lim, A. S., Melrose, Z. R., Thostenson, E. T., & Chou, T. W. (2011). Damage sensing of adhesively-bonded hybrid composite/steel joints using carbon nanotubes. Composites Science and Technology71(9), 1183-1189.

[b] Sam-Daliri, O., Faller, L. M., Farahani, M., Roshanghias, A., Araee, A., Baniassadi, M., ... & Zangl, H. (2019). Impedance analysis for condition monitoring of single lap CNT-epoxy adhesive joint. International Journal of Adhesion and Adhesives88, 59-65.

5.       In reference [14], the authors mentioned thermoplastic polyurethane/graphene strain sensor. Can you describe here the method of fabrication and how the graphene material dispersed within the polyurethane thermoplastic material? This is important for fabrication of thermoplastic nanocomposite as a piesoresitive material. There are some methods such as material extrusion filament which can help to have an even distribution of short fibre within the thermoplastic material. This fabrication method has investigated in some studies as the following which can be added to your study:

[c] Ghabezi, P., & Harrison, N. (2020). Mechanical behavior and long-term life prediction of carbon/epoxy and glass/epoxy composite laminates under artificial seawater environment. Materials Letters261, 127091.

6.       In the process of developing materials for sensor preparation, how did you determine the optimal percentage of MWCNTs and dispersion time to identify the percolation threshold? This is necessary as the conductivity mechanism can be changed from tunnelling effect to contact resistance.

7.       Why does joule heating contribute to the increased compressive resistance? Explore the mechanical and chemical interactions occurring between the nano-filler (CNT) and thermoplastic matrix that lead to this enhancement.

Author Response

(The authors gave the same response as above.)

Round 2

Reviewer 3 Report

Comments and Suggestions for Authors

 Accept in present form